# Artificial intelligence-based modeling for accurate leaf area estimation in olive (Olea europaea L.) cultivars

Yazgan Tunç[1], Fatih Demirel[2*], Ali Khadivi[3*], Kadir Uğurtan Yılmaz[4], Hüsnü Demirsoy[5], Bilal Cemek[6], Hatice Gözel[7], Daya Shankar Mishra[8]

**1** Republic of Türkiye, Ministry of Agriculture and Forestry, General Directorate of Agricultural Research and Policies, Hatay Olive Research Institute Directorate, Hassa, Hatay, Türkiye, **2** Department of Agricultural Biotechnology, Faculty of Agriculture, Igdir University, Iğdır, Türkiye, **3** Department of Horticultural Sciences, Faculty of Agriculture and Natural Resources, Arak University, Arak, Iran, **4** Department of Horticulture, Faculty of Agriculture, Kahramanmaraş Sütçü İmam University, Onikisubat, Kahramanmaraş, Türkiye, **5** Department of Horticulture, Faculty of Agriculture, Ondokuz Mayıs University, Samsun, Türkiye, **6** Department of Agricultural Structures and Irrigation, Faculty of Agriculture, Ondokuz Mayıs University, Atakum, Samsun, Türkiye, **7** Department of Horticulture, Faculty of Agriculture, Kilis 7 Aralık University, Kilis, Türkiye, **8** Department of Fruit Sciences, ICAR-CIAH RS Central Horticultural Experiment Station, Godhra, Gujarat, India

* fatih.demirel@igdir.edu.tr (FD); a-khadivi@araku.ac.ir (AK)

**Editor:** Marzia Vergine, University of Salento Department of Biological and Environmental Sciences and Technologies: Universita del Salento Dipartimento di Scienze e Tecnologie Biologiche ed Ambientali, ITALY

## Abstract

Estimating olive (Olea europaea L.) leaf area is an important aspect of monitoring plant health and evaluating growth processes in agriculture. Accurate estimation of leaf area allows for a better understanding of processes such as water and nutrient utilization, photosynthesis efficiency, respiration, and yield potential. This study aims to determine the most accurate, easy, and reliable leaf area estimation model using the geometric properties (length and width) of olive leaves. Additionally, the predictive performances of multiple linear regression (MLR) and artificial neural network (ANN) were compared. A total of 1320 leaf samples collected from 22 olive cultivars were used in the study. Leaf length and width were taken as input parameters, and both MLR and ANN models were developed for each cultivar. Both multiple linear regression (MLR) and artificial neural network (ANN) models demonstrated high predictive accuracy for olive leaf area estimation across 22 cultivars. The MLR models explained up to 96% of the variation in leaf area using leaf length (LL) and leaf width (LW), with low root mean square errors, indicating strong reliability. When cultivar identity was modeled as a categorical factor through dummy encoding, the model captured significant cultivar-specific effects without altering the overall predictive performance. The ANN models achieved slightly higher accuracy, with determination coefficients exceeding 0.99 and minimal prediction errors, confirming their superior ability to model nonlinear relationships. Across both approaches, leaf width contributed more strongly to leaf area than leaf length. Cultivar-specific differences were statistically significant for only a few genotypes, while most cultivars exhibited

**Data availability statement:** All relevant data are within the paper and its Supporting information files.

**Funding:** The author(s) received no specific funding for this work.

**Competing interests:** he authors have declared that no competing interests exist.

comparable patterns after adjustment for multiple testing. In conclusion, both MLR and ANN models demonstrated high accuracy in predicting olive leaf area, with ANN models showing slightly superior performance. However, MLR models also yielded highly reliable results, indicating that both approaches are viable for practical applications in olive cultivation. These predictive models can be effectively used for rapid, non-destructive phenotyping, growth monitoring, and precision management in olive breeding and production systems.

## 1. Introduction

The olive tree (Olea europaea L.) is one of the most widely cultivated fruit trees worldwide, particularly well-suited to the Mediterranean climate [1]. Various olive cultivars are grown depending on their intended use (edible olives and/or olive oil) and local microclimatic conditions [2,3].

With a cultivation history that spans over three thousand years, the olive tree is considered to be one of the most important and ancient fruit crops in the world [4]. It is especially well-suited to the Mediterranean environment, which is characterized by hot, dry summers and warm, rainy winters, which are favorable for its development. Olive is a significant crop in countries such as Spain, Italy, Greece, Türkiye, and Tunisia, as over ninety-five percent of the world's olive production is located in the Mediterranean basin [5]. Olive growth has spread beyond the Mediterranean area to other regions of the globe, including North and South America, Australia, and China, as a result of the economic and ecological advantages that olive farming offers [6]. An olive tree's capacity to flourish in dry and semi-arid settings, displaying a high tolerance to water shortages and poor soils, is one of the reasons why olive trees are so highly appreciated. Because of their extensive root systems, they can draw water from the deeper layers of the soil, which makes them an essential species for sustainable agriculture in regions that are prone to drought [7].

Estimation of leaf area (LA) in plants is considered a critical parameter in many scientific fields such as plant growth, agricultural productivity, ecosystem dynamics, and climate change studies [8–10]. Leaf area is a fundamental growth indicator that directly affects the photosynthetic capacity, transpiration rates, and biomass of the plant [11]. Leaf area is directly related to photosynthetic efficiency and growth rates. Leaf area width determines biomass increase by affecting the light capture capacity and carbon assimilation rates of the plant [12]. Plant parameters such as specific leaf area (SLA) and leaf mass area (LMA) provide information about water use, drought tolerance, and growth rate of the plant [13]. Leaf area is an important indicator for yield estimation and plant health monitoring in agricultural production. Leaf area width can lead to higher crop production by increasing photosynthetic efficiency. Furthermore, accurate calculation of leaf area index (LAI) in plants allows optimization of irrigation and fertilization strategies [14]. Plant water use and evaporation-transpiration processes depend on leaf area. Plants with large leaf areas may lose more water, while species with small leaf areas may be more tolerant of drought conditions. Thus,

leaf area measurements are used in planning agricultural irrigation programs and improving water use efficiency [15]. On the other hand, a decrease in leaf area may be a sign of disease, pest infestation, nutrient deficiency, or environmental stresses. Monitoring the change in leaf area over time is an important tool for the early detection of plant diseases and the development of intervention strategies [8]. In ecosystem studies, the leaf area index (LAI) is used to determine how much carbon plants can sequester from the atmosphere and how much biomass they can create through photosynthesis. In forest ecosystems, LAI calculations are a critical variable for modeling the carbon cycle and the effects of climate change [16,17]. Leaf area estimation is a vital variable in many areas such as plant growth, agricultural productivity, water management, plant health, ecosystem dynamics, and precision agriculture practices. Scientific studies show that accurate calculation of leaf area index improves agricultural management, contributes to climate change models, and is critical to understanding ecological processes [18].

In recent years, innovative technologies such as hand-held scanners and laser optical devices have become widely used in leaf area (LA) measurements. Although these advanced technologies offer high accuracy, they are generally preferred for more specialized research rather than basic experiments due to their complex structures and high costs [19]. In contrast, indirect methods offer significant advantages such as lower cost, speed, practicality, and less labor requirement compared with direct measurements. In addition, they are widely used in agricultural research and ecophysiological studies, enabling precise estimations without damaging the plant [20].

Indirect methods are quite valuable for investigating the empirical relationships between leaf area (LA) estimation and leaf dimensions. In particular, lamina dimensions such as leaf length (L) and leaf width (W) are widely used variables in estimating LA [21,22]. Various studies have shown that linear leaf measurements correlate strongly with LA. In this direction, studies have been conducted on leaf area estimation in many fruit trees, such as apple [23–25], pear [26], peach [27], apricot [28], grape [29], strawberry [30], cherry [31], citrus [32], chestnut [33], hazelnut [34], kiwi [35], black walnut [36], walnut [20], and mango [37].

Nowadays, there is a growing trend of utilizing machine learning (ML) techniques to address intricate issues. ML has been acquiring popularity at a rapid pace as a result of its capacity to resolve challenges in a variety of scientific research disciplines [38–41]. The importance of ML is derived from its capacity to analyze intricate datasets, reveal concealed patterns, and produce exact predictions. ML algorithms are a potent instrument in a variety of fields due to their superior predictive accuracy in comparison to traditional methods. Machine learning has become a fundamental component of contemporary problem-solving due to its extensive adaptability [42,43].

In recent years, ML's impact on agronomic applications has gained increasing attention. ML techniques have been successfully applied in various areas, including prediction of crop water content [44], estimation of crop nutrient concentration [45], modeling of crop yield [46], crop evapotranspiration estimation [46], and determination of soil moisture content [47]. To date, several ML-based, non-destructive models have been developed to estimate leaf area (LA) for different crop types. For example, artificial neural networks (ANNs) have proven to be successful in predicting LA for pepper [48], pear [26], tomato [49], durian [21], paprika [50], sugar beet [51], maize [52], cotton [53], and pepper [54].

Similarly, the Adaptive Neuro-Fuzzy Inference System (ANFIS) has also proven to be an effective method for LA estimation [55–57]. A comparative study by Sabouri et al. [58] demonstrated that ANN and ANFIS models outperformed traditional methods, such as multiple linear regression (MLR) and support vector regression (SVR), in LA estimation for plum genotypes. Specifically, the ANN and ANFIS models exhibited higher accuracy rates in leaf area estimation compared with conventional techniques.

This study aims to develop and compare LA estimation models for 22 widely cultivated olive cultivars in Türkiye using MLR and ANN approaches. Equations based on leaf length (LL), leaf width (LW), and the number of cultivars (CN) were formulated for both methods. Ultimately, this study aims to determine the most suitable approach for olive leaf area estimation and enhance its applicability in agricultural practices.

## 2. Materials and methods

### 2.1. Plant material and dataset

This study was conducted to develop and validate a leaf area estimation model for 22 olive cultivars grown in Türkiye (Republic of Türkiye, Ministry of Agriculture and Forestry, General Directorate of Agricultural Research and Policies, Hatay Olive Research Institute, Hassa Station) (Fig 1). The map was generated using ArcGIS software version 10.1 [59]. A total of 60 leaf samples were collected from each cultivar, and individual descriptive statistical characteristics are provided in S1 Table. All leaf samples were collected in January 2024. The leaves were collected from olive trees that were eight to ten years old. The dataset was split into training (70%) and testing (30%) subsets to ensure effective model development [60]. Table 1 presents the descriptive statistics for all olive cultivars, as well as the training and testing sets.

**Statement specifying permissions.** For this study, we acquired permission to study olive issued by the Agricultural and Forestry Ministry of the Republic of Türkiye.

**Statement on experimental research and field studies on plants.** The either cultivated or wild-growing plants sampled comply with relevant institutional, national, and international guidelines and domestic legislation of Türkiye.

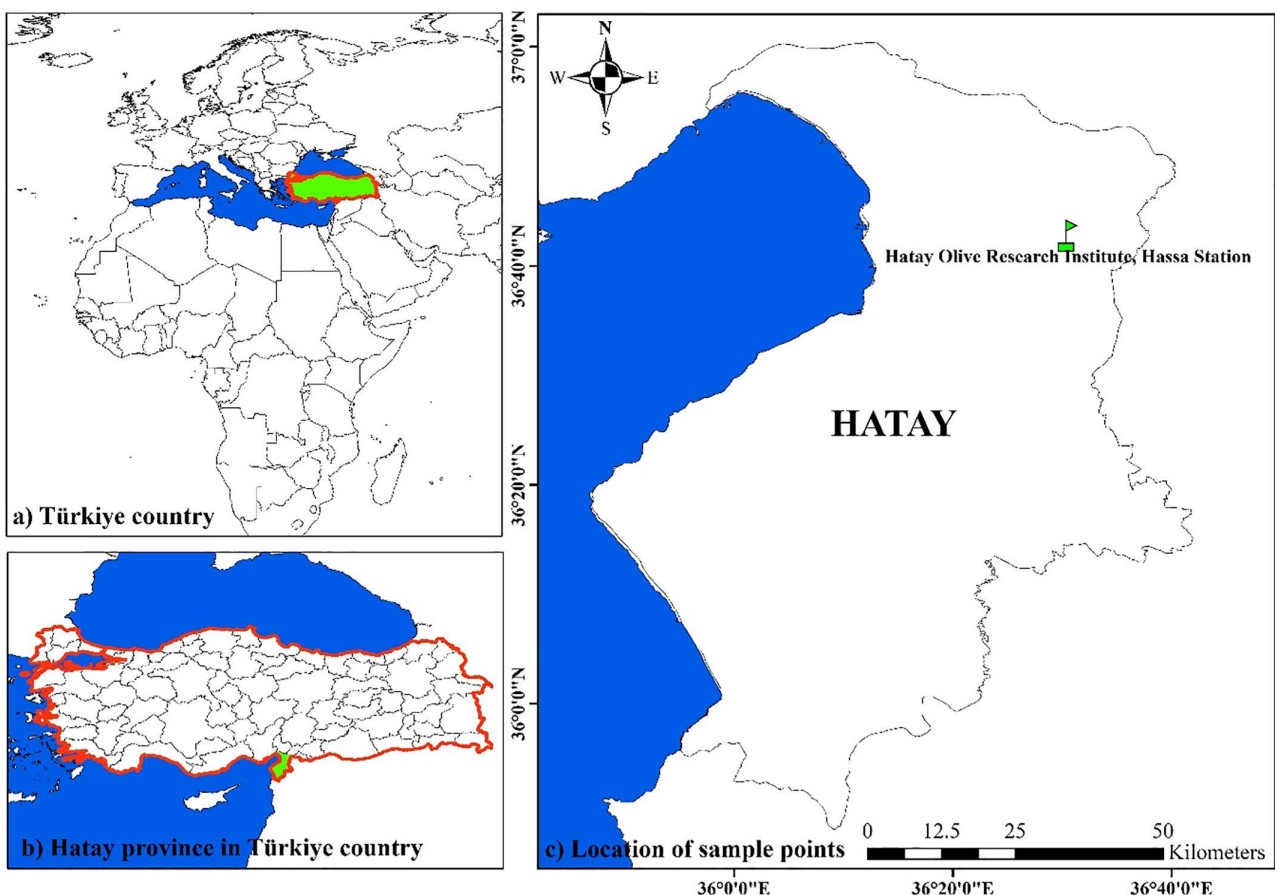

**Fig 1. Geographic location of the studied olive cultivars.** The above map image is original and was generated by the third author, Ali Khadivi, using ArcGIS software version 10.1 (http://www.arcgis.com/home/item.html?id=30e5fe3149c34df1ba922e6f5bbf808f).

**Table 1. Descriptive statistics for leaves of olive cultivars used in this study.**

| | Training (n = 880) | | | Testing (n = 440) | | | All (n = 1320) | | |
|---|---|---|---|---|---|---|---|---|---|
| | LL (mm) | LW (mm) | LA (cm2) | LL (mm) | LW (mm) | LA (cm2) | LL (mm) | LW (mm) | LA (cm2) |
| Min. | 12.02 | 4.53 | 0.56 | 12.52 | 5.31 | 0.56 | 12.02 | 4.53 | 0.56 |
| Max. | 93.81 | 29.78 | 14.58 | 68.94 | 27.37 | 9.12 | 93.81 | 29.78 | 14.58 |
| Avg. | 49.40 | 13.72 | 5.30 | 34.29 | 10.18 | 2.72 | 44.83 | 12.65 | 4.52 |
| SE | 0.53 | 0.13 | 0.09 | 0.54 | 0.15 | 0.07 | 0.45 | 0.11 | 0.07 |
| SD | 16.07 | 4.09 | 2.75 | 10.78 | 3.08 | 1.44 | 16.23 | 4.15 | 2.70 |
| Cv | 0.33 | 0.30 | 0.52 | 0.31 | 0.30 | 0.53 | 0.36 | 0.33 | 0.60 |
| Skewness | −0.49 | 0.54 | −0.17 | −0.36 | 4.09 | 2.75 | −0.30 | 0.64 | 0.29 |
| Kurtosis | 0.27 | 0.59 | 0.63 | 0.25 | 1.43 | 1.34 | 0.45 | 0.75 | 0.91 |

Min minimum, Max maximum, Avg average, SE standard error, SD standard deviation, Cv coefficient of variation, LA leaf area, LW leaf width, LL leaf length.

## 2.2. Leaf measurements

Each leaf sample was placed on an A4 sheet and photocopied at a 1:1 scale to preserve the actual size. The actual leaf area (LA) was determined using a Placom digital planimeter (Sokkisha Planimeter Inc., Model KP-90) from the photo-copied images. Additionally, leaf width (LW) and leaf length (LL) were measured for model construction. Leaf width (mm) was recorded at the widest part of the lamina from edge to edge, while leaf length (mm) was measured from the tip of the lamina to the petiole junction along the midrib. All measurements were taken to the nearest 0.1 cm for accuracy.

## 2.3. K-Fold cross-validation

All input and output variables were normalized to the range of 0–1 before training to ensure model convergence (see S1 Note for normalization equation Eq. 1).

K-fold cross-validation is a widely used technique for model validation, where the dataset is divided into K subsets (folds) to ensure robust evaluation across different partitions. In this study, 5-fold cross-validation was applied, with the dataset split into training (70%) and testing (30%) subsets (Fig 2). This approach helps prevent overfitting and improves the generalization of the model [61].

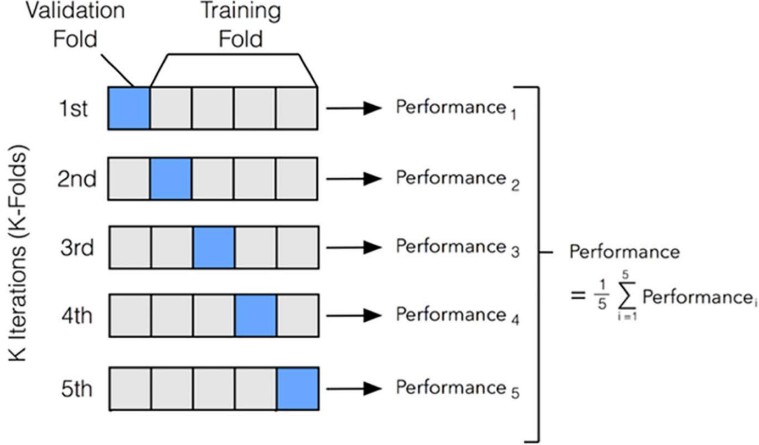

**Fig 2. K-fold cross-validation [61].**

## 2.4. Multiple linear regression (MLR)

Multiple linear regression (MLR) was employed to develop an analytical model for leaf area estimation using observational data from the 22 olive cultivars. Microsoft Office Excel 2015 (Microsoft Corporation, Redmond, WA, ABD) (https://www.microsoft.com) was used for regression analysis, incorporating different combinations of independent variables, primarily leaf length (LL) and leaf width (LW). The regression model was refined iteratively until the sum of squared deviations was minimized.

MLR is one of the most commonly used methods for leaf area estimation. A stepwise regression approach was utilized to determine the best-fit model. The independent variables were selected sequentially using backward stepwise, linear, and quadratic methods. The general form of the regression formula is given as Eq. 1.

$$y = b_0 + b_1\,X_1 + b_2\,X_2 \tag{1}$$

Where Y represents the dependent variable (leaf area, LA), b_0 is the intercept, b_1 and b_2 are regression coefficients, and X_1 and X_2 correspond to leaf width (LW) and leaf length (LL), respectively.

When two or more independent variables in the model (LL and LW) exhibit a high degree of correlation, collinearity may occur. Collinearity introduces redundancy in the information about the response variable, potentially diminishing the model's accuracy. To assess collinearity, variance inflation factors (VIFs) were computed for each predictor variable [62]. If the VIF values are less than 10, then there is no collinearity problem in the model. The VIF was calculated using Eq. 2.

$$[\text{VIF}]_j = 1/\left(1 - R_j^2\right) \tag{2}$$

Where represents the coefficient of determination for the regression model that includes all predictors except the jth predictor.

The cultivar number (CN) variable was defined as a categorical factor. To account for the fixed effects of cultivars, $K-1$ dummy (one-hot) indicator variables were created using treatment contrast coding, with 'Gemlik' selected as the reference cultivar. Accordingly, the general linear model was expressed as:

$$LA = \beta_0 + \beta_1 LL + \beta_2 LW + \Sigma(\gamma g \times Dg) \tag{3}$$

Where Dg represents the dummy variable corresponding to each cultivar (e.g., DAyv for 'Ayvalık', DDom for 'Domat', DMem for 'Memecik', etc.). By using this approach, the categorical nature of the cultivar factor was modeled correctly. The estimated dummy coefficients and their statistical significance levels for each cultivar are presented in S2 Table. The Table lists the estimated coefficients (γg), standard errors, and p-values for each cultivar dummy variable in Eq. 2. Positive coefficients indicate a higher mean leaf area than the reference cultivar ('Gemlik') at fixed LL and LW values, whereas negative coefficients indicate lower mean leaf area. This categorical treatment of cultivar identity using dummy (one-hot) encoding followed the procedure described by James et al. [63].

## 2.5. Multi-layer perceptron (MLP)

Multi-layer perceptron (MLP) is one of the most widely used artificial neural network (ANN) models. MLP consists of a network of interconnected units (neurons or nodes). These nodes operate as functions where the sum of incoming inputs is modified through a nonlinear transfer or activation function [64,65]. Connections are weighted, and each node generates an output signal.

In this study, different learning algorithms of artificial neural networks (ANNs) were tested for leaf area estimation, including Levenberg-Marquardt (LM), scaled conjugate gradient (SCG), and resilient backpropagation (RP) [66]. The

activation functions used in the model were Tansig (hyperbolic tangent sigmoid) between the input and hidden layers and Purelin (linear) between the hidden and output layers.

In Model 1, leaf width (LW) and leaf length (LL) were used as input variables, while leaf area (LA) was estimated as the output variable. In Model 2, LW, LL, and cultivar number (CN) were used as input variables, while LA was estimated as the output variable (Fig 3).

## 2.6. Model training and performance evaluation

The ANN models were trained using a backpropagation algorithm with a predefined number of epochs to ensure convergence. The performance of the models was evaluated using standard statistical metrics [67–69], including mean bias error (MBE), root mean squared error (RMSE), mean absolute error (MAE), and coefficient of determination ($R^2$). These metrics were used to compare ANN models with MLR, determining which model provided the most accurate leaf area estimation. Analyses were performed using MATLAB R2010b (http://www.mathworks.com/) software (The MathWorks Inc., Natick, MA). The MATLAB codes are provided as S1 Code.

To statistically compare the predictive performances of the models, a paired-sample t-test was performed between the observed, MLR-predicted, and ANN-predicted leaf area (LA) values for each olive cultivar as well as for the overall dataset (S3 Table). The analysis was used to test whether the prediction errors (residuals) of the ANN and MLR models differed significantly. For all cultivars, the test evaluated three paired groups: (i) observed vs. MLR, (ii) observed vs. ANN, and (iii) MLR vs. ANN. Statistical significance was determined at $p < 0.05$.

To further assess model generalization and potential overfitting, training loss convergence curves were recorded for both ANN architectures (2–3–1 and 3–4–1), as shown in S1 and S2 Figs. Each network was trained using the Levenberg–Marquardt optimization algorithm with a learning rate of 0.01 and an epoch range of 100–500. Early stopping was applied empirically at the iteration corresponding to the minimum root mean square error (RMSE), ensuring that the model did not overfit the training data.

## 2.7. ANN model development

The architectures and hyperparameter configurations of the ANN models (2–3–1 and 3–4–1) are summarized in S4 Table, including the number of hidden neurons, activation functions, learning algorithm, and epoch settings.

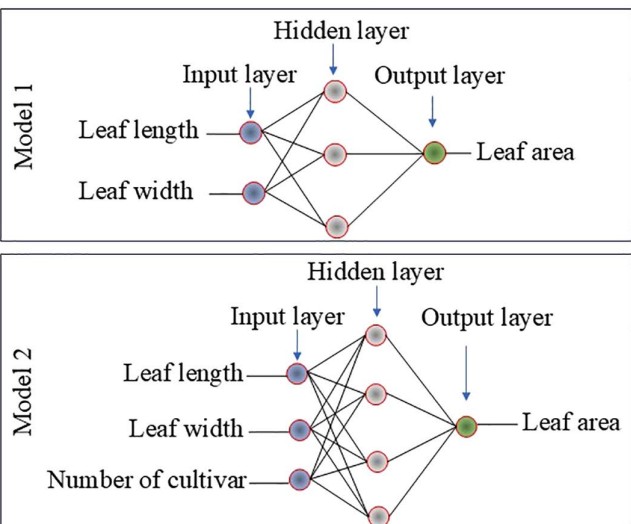

**Fig 3. Architecture of the multi-layer perceptron applied to estimate leaf area.**

## 3. Results

Descriptive statistical data for leaf area (LA), leaf length (LL), and leaf width (LW) of the olive cultivars, along with the training and test datasets, are presented in Table 1. In terms of leaf length (LL), the average values in the training dataset (49.40 mm) were higher than those in the test dataset (34.29 mm). This showed that the training dataset covered a wider range of leaf lengths. In terms of leaf width (LW), the mean values in the training dataset (13.72 mm) were higher than those in the test dataset (10.18 mm), indicating that the training dataset covered a wider range of leaf widths. For leaf area (LA), the average LA values in the test dataset (2.72 cm²) were lower than those in the training dataset (5.30 cm²). This indicated that smaller leaves were prevalent in the test dataset. The coefficient of variation (CV) value of leaf area was higher for the test dataset (0.53), while the leaf length value was higher for the training dataset (0.33). This indicated that leaf area exhibited more variability for the test dataset, while leaf length exhibited more variability for the training dataset. The coefficient of variation (0.30) value of leaf width was the same for the test and train datasets.

Leaf length (LL) was skewed to the left in both data sets (−0.49 for train and −0.36 for test), indicating slightly longer leaves than shorter ones. Leaf width (LW) had a highly skewed distribution in the test set (0.30), indicating some vast leaves compared with the rest. Leaf area (LA) was more symmetrical in the training set (−0.17) but skewed to the right in the test set, indicating that most leaves were small for the test set (2.75), with a few large ones, skewing the distribution to the right.

All LL, LW, and LA values were below 3, meaning that none of the distributions had strong peaks or extreme outliers. Leaf width (1.43) and leaf area (1.34) in the test set had higher kurtosis than in the train set (0.59 and 0.63, respectively), indicating more central clustering but still not extreme. Lower kurtosis values indicate a more even spread of the data rather than a high concentration near the mean.

### 3.1. Multiple linear regression (MLR) models

Before developing the LA model for different olive cultivars, collinearity between L and W was evaluated. Since the VIF values were less than 10, there was no collinearity between L and W. Therefore, these variables can be included in the LA models. The best MLR models for different olive cultivars are presented in S5 Table. For each olive cultivar, leaf area (LA) was estimated using leaf length (LL) and leaf width (LW) as independent variables.

In the training set, $R^2$ values generally ranged between 0.91 and 0.99, indicating that the models explained 91–99% of the variability in the training data. The $R^2$ values in the test set were generally between 0.92 and 0.99, showing a performance close to that of the training set. This indicates that the models have a high generalization ability. The MAE values in the training set typically ranged between 0.07 and 0.48, while in the test set, they varied between 0.11 and 1.52, demonstrating that the models had minor errors. The RMSE values in the training set generally ranged between 0.10 and 0.58, whereas in the test set, RMSE values were between 0.14 and 0.90. This suggests that the models kept even significant errors at a reasonable level.

When evaluating the model performance metrics ($R^2$, MAE, RMSE), all models yielded highly successful results. The training and test sets showed overall consistency, demonstrating strong generalization capability in the models. From a total of 22 olive cultivars, the Saurani cultivar had the best-performing model in the train set (R2: 0.99, MAE: 0.08, RMSE: 0.12), while the Girit Zeytini cultivar had the top-performing models in the test set (R2: 0.99, MAE: 0.14, RMSE: 0.21).

The coefficients in the models were statistically significant, indicating that leaf length and leaf width were significant factors in increasing leaf area. LL coefficients varied between 0.077 and 0.138, while LW coefficients were between 0.207 and 0.478. LW coefficients were higher than LL coefficients, suggesting that leaf width contributed more to leaf area.

Table 2 presents the general multiple linear regression (MLR) models developed for olive cultivars. The first model included only leaf length (LL) and leaf width (LW) as predictors, while the second model incorporated cultivar-specific dummy variables ($D_i$) representing each olive cultivar (e.g., D_Ayv for 'Ayvalık', D_Dom for 'Domat', D_Mem for

**Table 2. Multiple linear regression general leaf area models were developed for olive cultivars.**

| Model | Train (n = 880) | | | | Test (n = 440) | | | |
|---|---|---|---|---|---|---|---|---|
| | R2 | MBE | MAE | RMSE | R2 | MBE | MAE | RMSE |
| LA = −4.67 + 0.105LL + 0.347LW | 0.96 | −0.01 | 0.41 | 0.56 | 0.96 | 0.14 | 0.40 | 0.50 |
| LA = −4.58 + 0.10LL + 0.35LW + γAyvDAyv + γDomD-Dom + γMemDMem + ………………….. + γusluDuslu | 0.96 | 0.13 | 0.40 | 0.58 | 0.96 | 0.21 | 0.39 | 0.49 |

LA leaf area (cm2), LW leaf width (cm), LL leaf length (cm), R2 coefficient of determination, MBE mean bias error, MAE mean absolute error, RMSE root mean square error.

'Memecik', etc.), with corresponding coefficients ($\gamma_i$) estimating the specific effect of each cultivar on leaf area. This approach allowed the model to account for inter-cultivar variability more explicitly than a single numeric "cultivar number" (CN) parameter. Both models achieved identical $R^2$ values (0.96) for the training and test datasets, indicating strong predictive power and high generalization ability. Inclusion of dummy variables slightly adjusted the bias (MBE) and error (MAE, RMSE) metrics but did not markedly improve overall performance, suggesting that LL and LW remain the primary determinants of olive leaf area.

Fig 4 shows the linear regression between the predicted values and the actual values for Model 1 and Model 2. In Model 1, the coefficient of determination ($R^2$) values were 0.9567 for the training dataset and 0.9613 for the test dataset. The regression equations show that the estimated values closely followed the measured values, indicating that LL and LW alone provided a strong predictive capability. When Model 2 included CN as an additional input, the $R^2$ values improved slightly to 0.9565 for training and 0.9623 for testing. Although the improvement was minimal, the inclusion of cultivar numbers helped refine the model's predictive accuracy.

Overall, the MLR models demonstrated a strong capability to estimate leaf area based on leaf dimensions. The slight difference between the two models suggests that while cultivar number contributes to prediction accuracy, leaf length and leaf width remain the determinants of leaf area estimation.

This study also evaluated cultivar-specific effects on leaf area by re-specifying the cultivar number (CN) as a categorical factor through dummy (one-hot) encoding, using 'Gemlik' as the reference cultivar. In this revised model, cultivar-specific intercept differences in leaf area were estimated after controlling for leaf length (LL) and leaf width (LW). Following the Holm–Bonferroni correction, statistically significant positive coefficients were determined for 'Domat', 'Edincik Su', 'Elmacık', and 'Sarı Haşebi', indicating that these cultivars had higher mean leaf area than 'Gemlik' at fixed LL and LW values. Conversely, significantly negative coefficients were observed for 'Çelebi', 'Nizip Yağlık', and 'Sarı Ulak', indicating a lower mean leaf area compared with 'Gemlik'. For the remaining cultivars, no statistically significant differences were detected once multiplicity was adjusted. These findings demonstrated that the cultivar effect on leaf area was primarily concentrated in a limited number of genotypes, whereas variations among other cultivars were not statistically supported. The estimated dummy coefficients and their significance levels are presented in S6 Table.

### 3.2. Artificial neural network (ANN) models

All models were analyzed starting with 100 iterations, increasing by 100 until reaching 500 iterations. The best results for all 22 models were obtained between 200 and 300 iterations. In all models, tansig was used as the activation function between the input and hidden layers, while purelin was used between the hidden and output layers. The training algorithm was Levenberg-Marquardt (LM). The Levenberg–Marquardt algorithm demonstrated stable convergence without overfitting, as shown by the RMSE reduction curves in S1 and S2 Figs.

For the general model created using the combined data of all cultivars based on length and width as inputs, the same activation function and training algorithm were used. The dataset was split into 70% for training and 30% for testing,

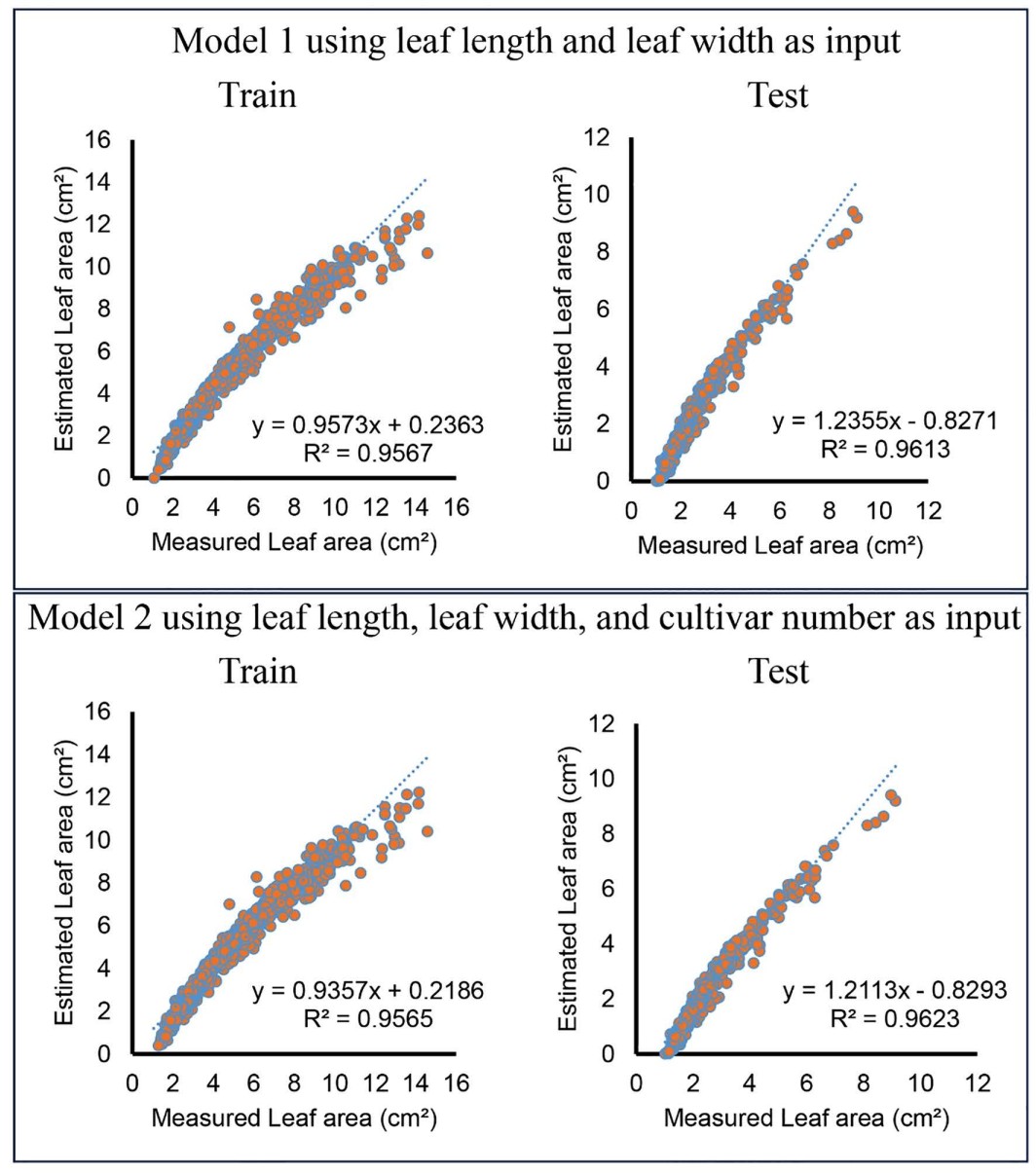

**Fig 4. The relationship between measured and estimated leaf area (LA) for the multiple linear regression (MLR) model in olive cultivars.**

with 880 samples for training and 440 for testing (Table 3). Another general model was developed by adding the cultivar number (CN) as an additional input, resulting in a three-input model with four neurons in the hidden layer. Different neuron numbers were tested in the hidden layers, and the best predictions were obtained using 2-3-1 and 3-4-1 network architectures. Additionally, different activation functions and training algorithms were tested, but the best results for all models were consistently achieved with tansig and purelin activation functions and the LM training algorithm.

The ANN models demonstrated high predictive performance for leaf area estimation across different olive cultivars. Two general models were developed: the first used leaf length (LL) and leaf width (LW) as input variables, while the second model incorporated cultivar number (CN) as an additional input (Table 4). For the first model, the ANN achieved an $R^2$

**Table 3. Estimation of the performance of the ANN in both training and testing phases for olive cultivar.**

| Cultivars | CN | Train | | | | Test | | | |
|---|---|---|---|---|---|---|---|---|---|
| | | MBE | MAE | RMSE | R2 | MBE | MAE | RMSE | R2 |
| 'Arbequina' | 1 | 0.050 | 0.140 | 0.019 | 0.998 | −0.043 | 0.051 | 0.077 | 0.990 |
| 'Ayvalık' | 2 | 0.001 | 0.173 | 0.028 | 0.990 | 0.034 | 0.083 | 0.119 | 0.989 |
| 'Çelebi' | 3 | −0.347 | 0.358 | 0.103 | 0.985 | −0.217 | 0.222 | 0.266 | 0.982 |
| 'Domat' | 4 | 1.045 | 1.045 | 0.775 | 0.989 | 0.324 | 0.324 | 0.384 | 0.977 |
| 'Edincik Su' | 5 | 0.461 | 0.501 | 0.193 | 0.994 | 0.236 | 0.297 | 0.374 | 0.949 |
| 'Elmacık' | 6 | 0.086 | 0.103 | 0.009 | 0.960 | 0.996 | 1.485 | 0.819 | 0.970 |
| 'Frantoio' | 7 | −0.214 | 0.360 | 0.100 | 0.982 | −0.219 | 0.222 | 0.268 | 0.994 |
| 'Gemlik' | 8 | −0.060 | 0.159 | 0.022 | 0.992 | −0.072 | 0.087 | 0.121 | 0.981 |
| 'Gemlik-21' | 9 | 0.037 | 0.179 | 0.028 | 0.991 | 0.042 | 0.149 | 0.199 | 0.975 |
| 'Girit Zeytini' | 10 | −0.085 | 0.098 | 0.007 | 0.988 | −0.093 | 0.093 | 0.118 | 0.997 |
| 'Halhalı' | 11 | 0.061 | 0.136 | 0.013 | 0.990 | 0.069 | 0.069 | 0.085 | 0.991 |
| 'Karamani' | 12 | −0.237 | 0.246 | 0.047 | 0.992 | −0.208 | 0.209 | 0.244 | 0.967 |
| 'Kilis Yağlık' | 13 | −0.340 | 0.373 | 0.130 | 0.988 | −0.128 | 0.138 | 0.172 | 0.961 |
| 'Manzanilla' | 14 | −0.183 | 0.224 | 0.035 | 0.992 | −0.192 | 0.195 | 0.225 | 0.969 |
| 'Memecik' | 15 | −0.030 | 0.174 | 0.026 | 0.983 | −0.009 | 0.068 | 0.075 | 0.992 |
| 'Nizip Yağlık' | 16 | −0.335 | 0.430 | 0.258 | 0.912 | −0.100 | 0.127 | 0.176 | 0.979 |
| 'Sarı Haşebi' | 17 | 0.143 | 0.156 | 0.027 | 0.996 | 0.012 | 0.093 | 0.117 | 0.931 |
| 'Sarı Ulak' | 18 | −0.254 | 0.286 | 0.052 | 0.990 | −0.217 | 0.220 | 0.257 | 0.995 |
| 'Sarı Yaprak' | 19 | 0.011 | 0.251 | 0.046 | 0.985 | 0.016 | 0.074 | 0.088 | 0.995 |
| 'Saurani' | 20 | 0.003 | 0.059 | 0.003 | 0.992 | 0.066 | 0.066 | 0.075 | 0.984 |
| 'Tavşan Yüreği' | 21 | 0.164 | 0.323 | 0.079 | 0.988 | 0.080 | 0.169 | 0.229 | 0.992 |
| 'Uslu' | 22 | 0.013 | 0.246 | 0.053 | 0.984 | −0.130 | 0.134 | 0.171 | 0.981 |

CN number of cultivar, R2 coefficient of determination, MBE mean bias error, MAE mean absolute error, RMSE root mean square error.

**Table 4. Estimation of the performance of ANN in both training and testing phases for olive cultivars.**

| All cultivars | Train | | | | Test | | | |
|---|---|---|---|---|---|---|---|---|
| | MBE | MAE | RMSE | R2 | MBE | MAE | RMSE | R2 |
| Model 1 | 0.001 | 0.275 | 0.432 | 0.975 | −0.052 | 0.166 | 0.227 | 0.975 |
| Model 2 | −0.001 | 0.167 | 0.245 | 0.992 | 0.019 | 0.142 | 0.040 | 0.995 |

Model 1: leaf length and leaf width as input; Model 2: leaf length, leaf width, and number of cultivars as input.

value of 0.975 in both the training and test sets, with an RMSE of 0.432 and 0.227, respectively. When the cultivar number was included as an input, model performance slightly improved, with the training $R^2$ increasing to 0.992 and RMSE reducing to 0.245. Among individual cultivars, the best-performing ANN models in the test set were observed for Girit Zeytini ($R^2$: 0.997, RMSE: 0.118, MAE: 0.093) and Arbequina ($R^2$: 0.990, RMSE: 0.077, MAE: 0.051), indicating highly reliable predictions (Table 3). However, the ANN models performed relatively less effectively for certain cultivars, such as Elmacık in the test set, where RMSE was 0.819, and MAE was 1.485, suggesting potential challenges in predicting leaf area for this cultivar (Table 3).

Overall, the ANN models exhibited strong predictive accuracy, with low MBE values, indicating minimal systematic bias. The inclusion of the cultivar number as an additional predictor slightly improved model performance, highlighting the potential influence of cultivar-specific traits on leaf area estimation.

## 3.3. ANN weight analysis

The ANN model's weight distributions between the input, hidden, and output layers are presented in Tables 6–8. The weight values indicate the significance of each feature in determining the final output.

## 3.4. Hidden layer weights (LL and LW as input)

Table 5 shows the weight values between the input layer and hidden layer for all olive cultivars. The connections between neurons varied significantly, with W1i values ranging from 0.523 to 0.0078 and W2i values ranging from 3.211 to 0.0232, suggesting that LW had a more substantial influence than LL. Bias values ranged from −6.979 to 16.960, which influenced the activation function shifts.

## 3.5. Output layer weights (LL and LW as input)

Table 6 presents the weight values between the hidden layer and the output layer. The final leaf area prediction was computed using a weighted sum of the hidden layer outputs, where weights varied significantly among neurons, particularly with Wi values ranging from 0.545 to 30.613. The bias value was 16.657, contributing to network adjustments.

The hidden layer outputs were calculated using the tansig activation function, defined as Eqs. 4–6.

$$[Output]_{11} = 2/\left(1 + \exp\left(-2 \times \left((0.523 \times LL) + (3.211 \times LW) - 6.979\right)\right)\right) - 1 \tag{4}$$

$$[Output]_{12} = 2/\left(1 + \exp\left(-2 \times \left((-0.255 \times LL) + (0.249 \times LW) + 16.960\right)\right)\right) - 1 \tag{5}$$

$$[Output]_{13} = 2/\left(1 + \exp\left(-2 \times \left((0.0078 \times LL) + (0.0232 \times LW) - 1.6677\right)\right)\right) - 1 \tag{6}$$

The final output was obtained using the purelin activation function (Eq. 7).

$$Output = (10.565 \times [Output]_{11}) + (0.545 \times [Output]_{12}) + (30.613 \times [Output]_{13}) + 16.657 \tag{7}$$

## 3.6. Hidden layer weights (LL, LW, and CN as input)

Table 7 presents the weight values when the cultivar number (CN) was included as an additional input. The weights indicate a more complex network structure, with W1i values ranging from 0.0159 to 0.0058, suggesting relatively minor contributions from LL. The bias values varied significantly (−1.628 to 5.309), altering activation function outputs.

**Table 5. The weight values between the input layer and the hidden layer for olive cultivars.**

| Weights | Number of neurons in the hidden layer (i) | | |
|---|---|---|---|
| | 1 | 2 | 3 |
| W1i | 0.523 | −0.255 | 0.0078 |
| W2i | 3.211 | 0.249 | 0.0232 |
| Bias1 | −6.979 | 16.960 | −1.6677 |

**Table 6. The weight values between the output layer and the hidden layer for olive cultivars.**

| Weights | Number of neurons in the hidden layer (i) | | | |
|---|---|---|---|---|
| | 1 | 2 | 3 | Bias 2 |
| Wi | 10.565 | 0.545 | 30.613 | 16.657 |

**Table 7. The weight values between the input layer and the hidden layer for olive cultivars.**

| Weights | Number of neurons in the hidden layer (i) | | | |
|---------|------|------|------|------|
| | **1** | **2** | **3** | **4** |
| W1i | 0.0159 | 0.640 | 0.0096 | 0.0058 |
| W2i | 0.0512 | 0.290 | 0.1615 | 0.16884 |
| W3i | −0.0017 | −0.307 | −0.0125 | −0.01383 |
| Bias1 | −1.628 | 5.309 | −4.627 | −4.55271 |

The complete weight matrices for the ANN model incorporating dummy-encoded cultivar variables (reference: 'Gemlik') are provided in S6 Table.

### 3.7. Output layer weights (LL, LW, and CN as input)

Table 8 displays the weight values between the hidden and output layers when CN was included. The output was computed using:

The hidden layer outputs were calculated using the tansig activation function, defined as Eqs. 8–11.

$$[\text{Output}]_{11} = 2/\left(1 + \exp\left(-2 \times ((0.0159 \times LL) + (0.0512 \times LW) + (-0.0017 \times CN) - 1.628)\right)\right) - 1 \tag{8}$$

$$[\text{Output}]_{12} = 2/\left(1 + \exp\left(-2 \times ((0.640 \times LL) + (0.290 \times LW) + (-0.307 \times CN) - 5.309)\right)\right) - 1 \tag{9}$$

$$[\text{Output}]_{13} = 2/\left(1 + \exp\left(-2 \times ((0.0096 \times LL) + (0.1615 \times LW) + (-0.0125 \times CN) - 4.627)\right)\right) - 1 \tag{10}$$

$$[\text{Output}]_{14} = 2/\left(1 + \exp\left(-2 \times ((0.0058 \times LL) + (0.16884 \times LW) + (-0.01383 \times CN) - 4.55271)\right)\right) - 1 \tag{11}$$

The final output was obtained using the purelin activation function (Eq. 12).

$$\text{Output} = (0.774 \times [\text{Output}]_{11}) + (-0.329 \times [\text{Output}]_{12}) + (-0.537 \times [\text{Output}]_{13}) + (-2.591 \times [\text{Output}]_{14}) + 1.868 \tag{12}$$

Overall, the ANN model demonstrated that LW had a more substantial effect on leaf area prediction than LL, and the inclusion of cultivar number slightly improved performance, as reflected in weight distributions and bias shifts. Since Model 2 also accounts for cultivar numbers, it provides a more detailed prediction, but the performance difference between the two models is quite slight. If the cultivar is known, Model 2 should be preferred; if the cultivar is unknown, Model 1 would be the more practical choice.

The detailed mathematical formulation of the forward pass and network equations is provided in S2 Note.

Fig 5 presents the relationship between measured and estimated leaf area values for the ANN models. The scatter plots illustrate the model's performance for both training and testing datasets. In the model utilizing LL and LW as inputs,

**Table 8. The weight values between the output layer and the hidden layer for olive cultivars.**

| Weights | Number of neurons in the hidden layer (i) | | | | |
|---------|------|------|------|------|------|
| | **1** | **2** | **3** | **4** | **Bias 2** |
| Wi | 0.774 | −0.329 | −0.537 | −2.591 | 1.868 |

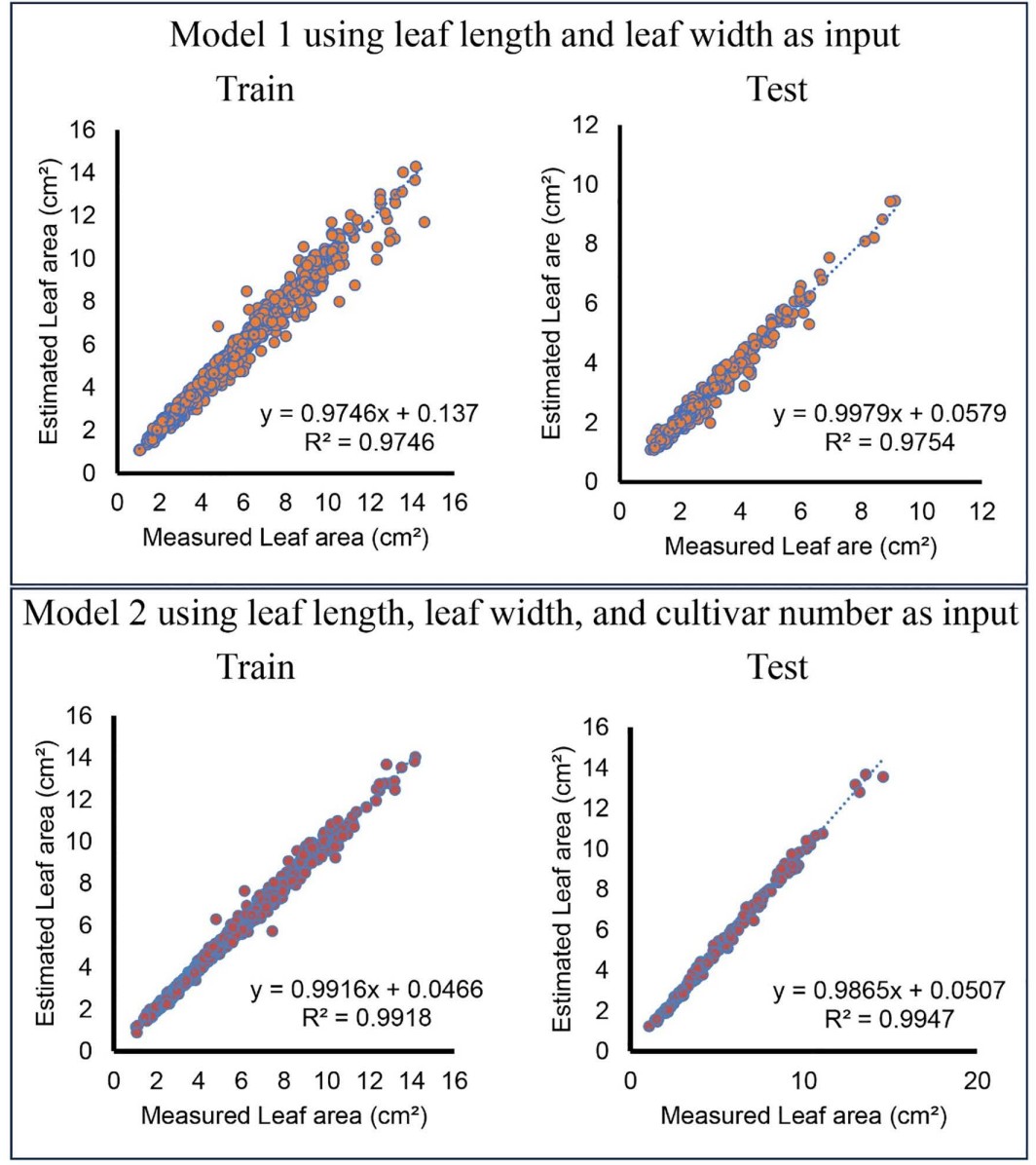

**Fig 5. The relationship between measured and estimated leaf area for the artificial neural networks (ANN) model in olive cultivars.**

the coefficient of determination ($R^2$) values were 0.9746 for the training set and 0.9754 for the test set, indicating strong agreement between estimated and measured values. When cultivar number (CN) was included as an additional input, the $R^2$ values improved slightly to 0.9918 for training and 0.9947 for testing, demonstrating that adding cultivar-specific information marginally enhanced the model's predictive power. The regression equations in the scatter plots showed a relationship close to 1:1 between the measured and predicted values, with slopes close to 1 and low intercepts. This indicates that the ANN model generalizes well across different olive cultivars and minimizes systematic biases. In general, the high $R^2$ values across all configurations indicate that the ANN model effectively captures the relationship between leaf

dimensions and leaf area. The slight improvement observed with the inclusion of CN indicated that cultivar-specific differences played a minor role in the estimation of leaf area.

## 4. Discussion

This study aimed to compare the performance of artificial neural networks (ANN) and multiple linear regression (MLR) models in predicting leaf area for different olive cultivars. Based on the obtained performance metrics ($R^2$, MAE, RMSE), both models produced successful results, but the ANN models exhibited higher accuracy.

Among the cultivars, 'Domat', 'Elmacık', and 'Sarı Haşebi' exhibited significantly higher mean leaf area values compared with the reference cultivar 'Gemlik' after adjustment for multiple testing. These differences likely reflect intrinsic morphological and physiological variations among olive genotypes. 'Domat', which is typically cultivated in humid and fertile regions of the Aegean Basin, is known for its vigorous growth habit and larger canopy structure, traits commonly associated with broader leaves and enhanced photosynthetic surface area. Similarly, 'Elmacık' is characterized by thick, elliptic leaves with high specific leaf area, which may confer improved photosynthetic efficiency under moderate water availability. In contrast, cultivars such as 'Çelebi' and 'Nizip Yağlık', adapted to drier environments, tend to possess smaller, narrower leaves as a xeromorphic adaptation to minimize transpiration. These findings indicate that the observed cultivar-specific deviations are not only statistical but also biologically consistent with the ecological and morphological diversity within Olea europaea germplasm. Comparable patterns have been reported in previous studies linking leaf morphological traits to environmental adaptation and genotypic variation in olives [2,3,7].

The overall success of the ANN model, attributed to its capacity to learn nonlinear relationships, demonstrated that it provides a more effective approach for leaf area estimation ($R^2 = 0.995$). In contrast, the coefficient of determination ($R^2$) for the MLR models was calculated as 0.96, both for the model using LL and LW as input parameters and for the model incorporating LL, LW, and CN as inputs. The inclusion of the CN variable in the ANN model improved performance and enhanced generalizability, enabling more comprehensive predictions. Overall, ANN models outperformed MLR models in both the training and testing phases.

Yuan et al. [70] compared different regression methods, including ANN, random forest (RF), and support vector machine (SVM) models, for predicting the leaf area index (LAI) in soybean crops. Their study concluded that ANN provided the highest precision, particularly for single growth phases, outperforming SVM and traditional statistical approaches. Similarly, Kumar et al. [71] developed an ANN model incorporating leaf length and width as inputs for durian leaf area estimation. They compared ANN with regression models and found that ANN yielded significantly better results, achieving an $R^2$ of 0.96 in the training phase and 0.94 in the testing phase, outperforming linear regression methods. These findings align with our study, where ANN-based models demonstrated a strong correlation between estimated and actual leaf area across different olive cultivars.

Shabani et al. [72] further validated ANN's capability by testing it on 61 plant species with varying leaf morphologies. Their study revealed that ANN models performed consistently well across different species, reinforcing their versatility and robustness. The research also highlighted the importance of selecting input parameters, showing that pre-processing methods, such as MinMax normalization, significantly improved ANN model accuracy. These findings suggest that ANN models are adaptable to different plant species and growth conditions, making them ideal for agricultural applications.

The findings of this study align with previous research on similar topics. Kumar et al. [71], Küçükönder et al. [73], and Ahmadian-Moghadam [54] also demonstrated that ANN models achieve high accuracy in leaf area estimation. However, Shabani et al. [72] emphasized the need to develop cultivar-specific models, as generalized ANN models may underperform for certain plant species.

Additionally, Kallel et al. [74] indicated that ANN models can be effectively utilized for leaf area index (LAI) estimation, proving beneficial for monitoring olive tree health. Similarly, Mhanna [75] reported that mathematical models developed for estimating leaf area in the 'Khoderi' olive cultivar in Syria achieved an $R^2$ value of 0.962, and the same approach was

successfully applied to the French 'Picholine' cultivar. This further supports the idea that cultivar-specific models yield more accurate predictions.

Koubouris et al. [76] highlighted that using both leaf length (LL) and width (LW) together for leaf area estimation produces better results than using either variable alone. Their modeling study on 10 different olive cultivars showed R² values ranging from 0.71 to 0.92. Similarly, in this study, using both LL and LW in ANN and MLR models improved prediction accuracy. In MLR models, leaf width (LW) was found to be more influential in predicting leaf area.

Öztürk et al. [26] confirmed that ANN models outperformed MLR models in predicting the leaf area (LA) of pear cultivars. They found that ANN models provided higher accuracy in both training and testing phases, with R² values reaching 0.991 in training and 0.985 in testing, while MLR models showed slightly lower performance with R² values of 0.987 in training and 0.979 in testing. These findings further validate the conclusions of our study, which demonstrated that ANN models offer a more reliable and accurate approach to LA estimation compared with traditional regression-based models.

In a study by Asriani [77], the performance of artificial neural networks (ANN) in leaf area estimation was examined using the backpropagation algorithm. Leaf length and width were used as input variables, while leaf area was the output variable. Their results showed that an ANN model with a 2-50-1 architecture achieved 99.99% accuracy in predicting leaf area for seven plant species.

Kishore et al. [78] developed a non-destructive model for estimating apple (Malus domestica) leaf area, using linear leaf measurements such as leaf length (L) and width (W). The study tested multiple regression models and found that the most accurate model was LA = 1.120301 + 0.615(L × W). This model achieved an R² of 0.98, demonstrating high accuracy in predicting leaf area across multiple apple cultivars. Kumar and Sharma [79] developed a leaf area estimation model for Picrorhiza kurroa, an important medicinal plant. They explored multiple linear, quadratic, exponential, and logarithmic regression models using leaf length (L) and width (W) as independent variables. The most accurate model was found to be LA = 0.333 + 0.603(L × W), which achieved an R² of 0.955 in model calibration and 0.9053 in validation, demonstrating high accuracy and robustness. These findings further support the idea that empirical models using leaf dimensions can provide reliable non-destructive leaf area estimation. However, our study, along with others incorporating artificial neural networks (ANNs) [26,71], suggests that machine learning-based approaches offer an even more precise and adaptable solution, particularly for complex and heterogeneous datasets.

Monitoring plant health and evaluating growth processes are crucial for enhancing agricultural productivity and achieving sustainable production goals [8]. In this context, leaf area estimation is a key parameter for assessing water and nutrient use, evaluating photosynthetic efficiency, and analyzing yield potential. Accurately and reliably estimating leaf area in economically significant crops such as olives contributes to a better understanding of plant physiology and supports strategic decision-making in agricultural practices.

The findings of this study indicate that ANN models offer a strong alternative for leaf area estimation, providing more accurate results compared with MLR models. However, it is concluded that cultivar-specific analyses and the integration of additional variables play a critical role in improving model performance.

The results show that ANN produces lower error without creating systematic bias compared to MLR with the same inputs. Effect sizes indicate small-to-medium/medium-level improvements in practical terms and quantitatively confirm an accuracy advantage consistent with high R² values.

## 5. Conclusions

In this study, multiple linear regression (MLR) and artificial neural network (ANN) models were developed to predict leaf area (LA) for 22 different olive cultivars. While leaf length (LL) and leaf width (LW) were used for individual predictions, cultivar number (CN) was included in general predictions. The developed models enabled non-destructive and cost-effective leaf area estimation, eliminating the need for expensive equipment. Deterministic models that can be easily used in accession were formulated.

Leaf area is a critical parameter for plant growth and productivity. Various methods exist for LA measurement, including regression models, grid counting, gravimetric analysis, planimetry, image processing, and adaptive neuro-fuzzy inference systems. In this study, ANN models successfully produced results using a 2-3-1 network architecture for individual predictions and a 3-4-1 network architecture for general predictions. Comparative analyses revealed that ANN models outperformed MLR models in LA prediction. ANNs demonstrated greater flexibility and adaptability to different olive cultivars, making them a more effective approach. However, MLR models remain a practical and reliable method due to their ease of use. Predicting olive leaf area is crucial for monitoring plant health and productivity in agriculture. AI-based methods can analyze satellite imagery data to assess plant health and stress levels. LA estimation can help optimize production planning and prevent fraud in the olive industry by providing early yield predictions.

In conclusion, AI-based methods are effective and innovative tools for predicting olive leaf area. The integration of artificial neural networks (ANN) with remote sensing technologies could further enhance agricultural productivity and support the development of sustainable farming practices. The findings of this study confirm that ANN models provide a strong alternative for leaf area estimation, offering more accurate results than MLR models. However, it is concluded that cultivar-specific analyses and the integration of additional variables play a critical role in improving model performance. Practically, the developed models can be applied as rapid, non-destructive, and cost-efficient tools for phenotypic characterization, growth monitoring, and precision management in olive breeding and cultivation systems. Future research should aim to validate these models under field conditions, expand datasets with more diverse genotypes and environmental factors, and explore advanced algorithms such as convolutional or ensemble learning methods to further improve predictive accuracy and generalization capability.

## Supporting information

**S1 Table. Descriptive statistics for individual olive cultivars used in this study.**
(DOCX)

**S2 Table. Dummy (one-hot) coefficients for cultivars (reference: 'Gemlik') in the multiple linear regression model.**
(DOCX)

**S3 Table. Paired-sample t-test results for comparison of LA estimation methods.**
(DOCX)

**S4 Table. Hyper-parameter and functions selected during the training phase.**
(DOCX)

**S5 Table. Multiple linear regression leaf area models were developed for olive cultivar.**
(DOCX)

**S6 Table. Weight values between the input and hidden layers of the ANN model incorporating dummy (one-hot) encoded cultivar variables (reference: 'Gemlik').**
(DOCX)

**S1 Note. Normalization of input and output data.**
(DOCX)

**S2 Note. Mathematical formulation of the ANN model.**
(DOCX)

**S1 Fig. Iteration RMSE change according to 2-3-1 network structure.**
(DOCX)

 

**S2 Fig. Iteration RMSE change according to 3-4-1 network structure.**
(DOCX)

**S1 Code. The MATLAB codes used.**
(DOCX)

## Author contributions

**Conceptualization:** Ali Khadivi, Hüsnü Demirsoy, Hatice Gözel.

**Data curation:** Kadir Uğurtan Yılmaz, Hüsnü Demirsoy, Hatice Gözel, Daya Shankar Mishra.

**Formal analysis:** Yazgan Tunç, Fatih Demirel, Kadir Uğurtan Yılmaz, Hüsnü Demirsoy, Bilal Cemek.

**Investigation:** Yazgan Tunç, Fatih Demirel, Ali Khadivi, Kadir Uğurtan Yılmaz, Hüsnü Demirsoy, Bilal Cemek.

**Methodology:** Yazgan Tunç, Ali Khadivi, Bilal Cemek, Daya Shankar Mishra.

**Project administration:** Hatice Gözel, Daya Shankar Mishra.

**Supervision:** Fatih Demirel.

**Validation:** Kadir Uğurtan Yılmaz.

**Writing – original draft:** Fatih Demirel, Ali Khadivi, Kadir Uğurtan Yılmaz, Hüsnü Demirsoy, Hatice Gözel, Daya Shankar Mishra.

**Writing – review & editing:** Yazgan Tunç, Ali Khadivi, Kadir Uğurtan Yılmaz, Hüsnü Demirsoy, Bilal Cemek.

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
