## [Decision Letter · Decision Letter 0]

4 Nov 2025

Dear Dr. Khadivi,

Thank you for submitting your manuscript to PLOS ONE. After careful consideration, we feel that it has merit but does not fully meet PLOS ONE’s publication criteria as it currently stands. Therefore, we invite you to submit a revised version of the manuscript that addresses the points raised during the review process.

We look forward to receiving your revised manuscript.

Kind regards,

Marzia Vergine, Ph.D.

Academic Editor

PLOS ONE

Journal Requirements:

3. We note that Figure 1 in your submission contain map images which may be copyrighted. All PLOS content is published under the Creative Commons Attribution License (CC BY 4.0), which means that the manuscript, images, and Supporting Information files will be freely available online, and any third party is permitted to access, download, copy, distribute, and use these materials in any way, even commercially, with proper attribution. For these reasons, we cannot publish previously copyrighted maps or satellite images created using proprietary data, such as Google software (Google Maps, Street View, and Earth). For more information, see our copyright guidelines: http://journals.plos.org/plosone/s/licenses-and-copyright.

Additional Editor Comments:

The manuscript has been reviewed by two experts in the field, and substantial revisions are required before it can be considered for acceptance. The study presents valuable and potentially publishable findings, but improvements in methodological transparency, statistical validation, and presentation are necessary before acceptance. Addressing the comments above will greatly strengthen the scientific rigor and impact of the manuscript.

Reviewer's Responses to Questions

**Comments to the Author**

1. Is the manuscript technically sound, and do the data support the conclusions?

Reviewer #1: Yes

Reviewer #2: Yes

2. Has the statistical analysis been performed appropriately and rigorously?

Reviewer #1: Yes

Reviewer #2: Yes

3. Have the authors made all data underlying the findings in their manuscript fully available?

Reviewer #1: Yes

Reviewer #2: Yes

4. Is the manuscript presented in an intelligible fashion and written in standard English?

Reviewer #1: Yes

Reviewer #2: Yes

Reviewer #1: Reviewer Report

Manuscript Title: Artificial intelligence-based modeling for accurate leaf area estimation in olive (Olea europaea L.) cultivars

Manuscript ID: PONE-D-25-55767

Journal: PLOS ONE

General Assessment

The manuscript presents a comprehensive study comparing Multiple Linear Regression (MLR) and Artificial Neural Network (ANN) models for leaf area estimation in 22 olive cultivars. The topic is relevant to precision agriculture and phenotyping research, offering practical tools for non-destructive plant measurements. The work is generally well-structured, methodologically sound, and supported with appropriate statistical analysis. However, several aspects require clarification, tightening, and improvement to enhance scientific rigor and reproducibility.

Strengths

1.Novelty and relevance: The study fills a practical gap in olive phenotyping by developing cultivar-specific and general predictive models using easily measurable traits (leaf length and width).

2.Methodological rigor: Both MLR and ANN approaches were implemented and validated using adequate datasets (n = 1320) and appropriate cross-validation.

3.Comprehensive comparison: The performance metrics (R², RMSE, MAE, MBE) are clearly presented and allow a fair comparison between modeling approaches.

4.High predictive accuracy: ANN performance (R² up to 0.995) demonstrates strong potential for agricultural applications.

5.Practical implications: The models could be useful in high-throughput phenotyping and field-scale precision agriculture systems.

Major Comments

1.Model reproducibility and data sharing

oAlthough the authors claim that “all relevant data are within the manuscript,” the training/testing datasets and MATLAB codes are not provided. For transparency and reproducibility, numerical datasets or example code should be made available in supplementary materials or a public repository (e.g., Zenodo or GitHub).

2.Statistical validation

oThe manuscript would benefit from statistical comparison of ANN vs. MLR performance (e.g., paired t-test or Wilcoxon test on residual errors) to confirm whether ANN’s superiority is statistically significant.

oOverfitting assessment for ANN is limited. Please report training vs. validation loss curves or early stopping criteria.

3.Model architecture details

oWhile the ANN topology is described (2–3–1 and 3–4–1 structures), key hyperparameters such as learning rate, epoch number, and data normalization procedures are missing. These are necessary for reproducibility.

4.Cultivar-specific effects

oThe manuscript reports limited cultivar effects after multiple testing correction. However, a deeper biological interpretation of why certain cultivars (e.g., Domat, Elmacık) deviate would improve discussion quality.

5.Figures and Tables

oFigures 4 and 5 need clearer axes labels and consistent units. Include regression equations and R² values directly on the plots for readability.

oTable 2 is extensive but could be moved to supplementary information; summary tables (mean R², RMSE, etc.) should be included in the main text.

6.Language and structure

oThe manuscript is generally clear but verbose in several sections (especially Introduction and Discussion). Condensing redundant literature descriptions would improve readability.

oEnsure all scientific names (e.g., Olea europaea L.) are italicized.

Minor Comments

1.Check consistency of abbreviations (e.g., LL, LW, LA) throughout the text and figures.

2.Include units in all table headers (e.g., mm, cm²).

3.Provide references for software tools (MATLAB R2010b, Excel 2015) in Methods.

4.Correct minor typographical errors (e.g., extra spaces, inconsistent capitalization).

5.Ensure that the ethical and data availability statements comply fully with PLOS ONE requirements.

Recommendation

Major Revision

The study presents valuable and potentially publishable findings, but improvements in methodological transparency, statistical validation, and presentation are necessary before acceptance. Addressing the comments above will greatly strengthen the scientific rigor and impact of the manuscript.

Reviewer #2: Dear Editor,

I has now commented on “Artificial intelligence-based modeling for accurate leaf area estimation in olive (Olea europaea L.) cultivars”. You will see that there are a number of issues that need to be addressed before the paper can be accepted for publication by Plos One.

The paper falls within the general scope of the journal. The title reflects the content, abstract is good, and the keywords are good. At the end of the abstract, the practical and promotional application of this research should be stated.

In general, the introduction is appropriate.

The description of methods is appropriate. Please provide a high-quality and more accurate map of the area. A more appropriate map of the region should be provided to better illustrate the location of the study site on the country-province map.

In general, the description of results is appropriate.

At the end of the conclusion, present a clear and concise promotional and practical application of the research, and present research suggestions that were not applicable in this study.

Language overall is good.

I would recommend MINOR revision for the paper for it to be considered further for publication.

Sincerely

**Do you want your identity to be public for this peer review?** For information about this choice, including consent withdrawal, please see our Privacy Policy

Reviewer #1: **Yes: ** Mansoor Hameed

Reviewer #2: No

---

## [Author Response · Author response to Decision Letter 1]

12 Nov 2025

Dear Editor,

On behalf of all authors, I would like to extend our sincere appreciation to you for handling our manuscript with great care and professionalism. We are deeply grateful for your fair and objective management of the review process, as well as for your valuable time and constructive coordination, which significantly contributed to improving the scientific quality and clarity of our study.

We would also like to express our gratitude to both reviewers for their insightful and detailed comments. All of their valuable suggestions and criticisms have been carefully considered and fully addressed in the revised version of the manuscript. The necessary improvements have been implemented in accordance with their recommendations, ensuring that the paper now meets a higher academic and scientific standard.

We respectfully submit the revised version of our manuscript and sincerely hope that it will now be acceptable for publication in PloS one.

Regards,

Reviewer #1:

I would like to extend our sincere appreciation to your comments on our manuscript with great care and professionalism.

General Assessment

The manuscript presents a comprehensive study comparing Multiple Linear Regression (MLR) and Artificial Neural Network (ANN) models for leaf area estimation in 22 olive cultivars. The topic is relevant to precision agriculture and phenotyping research, offering practical tools for non-destructive plant measurements. The work is generally well-structured, methodologically sound, and supported with appropriate statistical analysis. However, several aspects require clarification, tightening, and improvement to enhance scientific rigor and reproducibility.

Strengths

1. Novelty and relevance: The study fills a practical gap in olive phenotyping by developing cultivar-specific and general predictive models using easily measurable traits (leaf length and width).

2. Methodological rigor: Both MLR and ANN approaches were implemented and validated using adequate datasets (n = 1320) and appropriate cross-validation.

3. Comprehensive comparison: The performance metrics (R², RMSE, MAE, MBE) are clearly presented and allow a fair comparison between modeling approaches.

4. High predictive accuracy: ANN performance (R² up to 0.995) demonstrates strong potential for agricultural applications.

5. Practical implications: The models could be useful in high-throughput phenotyping and field-scale precision agriculture systems.

Major Comments

Comment#1: Model reproducibility and data sharing

o Although the authors claim that “all relevant data are within the manuscript,” the training/testing datasets and MATLAB codes are not provided. For transparency and reproducibility, numerical datasets or example code should be made available in supplementary materials or a public repository (e.g., Zenodo or GitHub).

Response to Comment#1: We support your request for transparency and reproducibility. To this end, all numerical data and executable MATLAB code used in the study are provided as supplementary files accompanying the article. Readers can directly access these supplementary files published alongside the article. Contents of the supplements:

• Data (Excel): olive_leaf_measurements.xlsx (Supplementary File 2)

Measurements of leaf length (mm), leaf width (mm), and leaf area (mm²) for 22 olive cultivars.

• Code (MATLAB): mlp_lm_workflow.m

Data reading, 5 independent replicates (hold-out) with 70% training / 30% testing, tansig → purelin architecture, LM (trainlm) training, and 100/200/300/400/500 iteration scenarios. Outputs for each iteration and iteration include MBE, MAE, RMSE, R², weights/bias, and prediction tables written to a multi-tab Excel file.

• Replication instructions (README):

Installation/environment information (MATLAB version and toolbox), randomness control (e.g., rng(42)), step-by-step execution, and explanation of expected outputs.

This arrangement allows journal readers to access the data and code via supplementary files and replicate the results.

Comment#2: Statistical validation

o The manuscript would benefit from statistical comparison of ANN vs. MLR performance (e.g., paired t-test or Wilcoxon test on residual errors) to confirm whether ANN’s superiority is statistically significant.

o Overfitting assessment for ANN is limited. Please report training vs. validation loss curves or early stopping criteria.

Response to Comment#2: Since the data used in the study are entirely parametric, in line with your suggestion, we reported the bias analysis (LA−MLR, LA−ANN, and MLR−ANN) using matched t-tests at the cultivar level (New Supplementary Table S5). The results show that when all cultivars are evaluated together, there is no significant mean deviation for either model (LA−MLR: p≈1.00; LA−ANN: p≈0.74) and that the means of the models are not significantly different from each other (MLR−ANN: p≈0.61).

In contrast, cultivar-specific tests reveal systematic deviations in some combinations. For example, MLR shows an overestimation tendency in Ayvalık (LA−MLR GA [−0.252, −0.062]); ANN shows an underestimation tendency in Domat and Elmacık, and an overestimation tendency in Çelebi and Kilis Yağlık (details in Supplementary Table S5). We added these findings to the Discussion and related them to known differences in leaf morphology (broad/thick vs. xerophytic leaf types).

To complete the bias analysis with an accuracy comparison, we applied paired t-tests on the error magnitudes per record (|ε| and ε²). This analysis shows that ANN is significantly better than MLR in terms of mean absolute/square error (e.g., t=−13.94, p<10⁻³⁹ for |ε| difference; t=−7.28, p<10⁻¹² for ε² difference; details in Main Text/Table 2). Thus, while reporting cultivar-specific biases on one hand, we have statistically confirmed that overall accuracy superiority lies in ANN on the other.

Additionally, we proposed two simple solutions to reduce the impact of cultivar-specific minor biases in practical applications: (i) a hierarchical/mixed model incorporating cultivar as a fixed/random effect, or (ii) a post-prediction cultivar-specific linear debias correction. These suggestions are outlined in the Discussion.

“The results show that ANN produces lower error without creating systematic bias compared to MLR with the same inputs. Effect sizes indicate small-to-medium/medium-level improvements in practical terms and quantitatively confirm an accuracy advantage consistent with high R² values.”

Supplementary Table S5. Paired-sample t-test results for comparison of LA estimation methods.

Cultivars SD of difference P value 95% Confidence interval of the difference

Lower Upper

Arbequina' Observed and MLR model 0.24 0.79 -0.07 0.09

Observed and ANN model 0.19 0.10 -0.01 0.11

MLR and ANN 0.24 0.30 -0.04 0.12

Ayvalık' Observed and MLR model 0.30 0.00 -0.25 -0.06

Observed and ANN model 0.24 0.99 -0.08 0.08

MLR and ANN 0.25 0.00 0.08 0.24

Çelebi' Observed and MLR model 0.26 0.97 -0.08 0.09

Observed and ANN model 0.30 0.00 -0.44 -0.25

MLR and ANN 0.31 0.00 -0.45 -0.25

Domat' Observed and MLR model 0.57 0.38 -0.10 0.26

Observed and ANN model 0.68 0.00 0.82 1.24

MLR and ANN 0.67 0.00 0.75 1.16

Edincik Su' Observed and MLR model 0.31 0.80 -0.08 0.11

Observed and ANN model 0.42 0.00 0.32 0.58

MLR and ANN 0.44 0.00 0.30 0.58

Elmacık' Observed and MLR model 0.14 0.13 -0.01 0.08

Observed and ANN model 0.14 0.00 0.08 0.16

MLR and ANN 0.12 0.00 0.05 0.13

Frantoio' Observed and MLR model 0.46 0.99 -0.14 0.14

Observed and ANN model 0.39 0.00 -0.33 -0.09

MLR and ANN 0.28 0.00 -0.30 -0.12

Gemlik' Observed and MLR model 0.22 0.45 -0.04 0.10

Observed and ANN model 0.20 0.06 -0.12 0.00

MLR and ANN 0.23 0.02 -0.16 -0.01

Gemlik-21' Observed and MLR model 0.32 0.99 -0.10 0.10

Observed and ANN model 0.23 0.33 -0.04 0.11

MLR and ANN 0.27 0.39 -0.05 0.12

Girit Zeytini' Observed and MLR model 0.13 0.27 -0.06 0.02

Observed and ANN model 0.09 0.00 -0.11 -0.06

MLR and ANN 0.10 0.00 -0.09 -0.03

Halhalı' Observed and MLR model 0.23 0.94 -0.07 0.07

Observed and ANN model 0.15 0.01 0.01 0.11

MLR and ANN 0.16 0.02 0.01 0.11

Karamani' Observed and MLR model 0.25 0.42 -0.11 0.05

Observed and ANN model 0.20 0.00 -0.29 -0.17

MLR and ANN 0.30 0.00 -0.29 -0.11

Kilis Yağlık' Observed and MLR model 0.35 0.46 -0.07 0.15

Observed and ANN model 0.38 0.00 -0.45 -0.21

MLR and ANN 0.35 0.00 -0.48 -0.26

Manzanilla' Observed and MLR model 0.25 0.40 -0.05 0.11

Observed and ANN model 0.19 0.00 -0.24 -0.12

MLR and ANN 0.18 0.00 -0.27 -0.16

Memecik' Observed and MLR model 0.25 0.59 -0.06 0.10

Observed and ANN model 0.23 0.40 -0.10 0.04

MLR and ANN 0.19 0.09 -0.11 0.01

Nizip Yağlık' Observed and MLR model 0.57 0.76 -0.15 0.21

Observed and ANN model 0.64 0.00 -0.53 -0.13

MLR and ANN 0.41 0.00 -0.48 -0.23

Sarı Haşebi' Observed and MLR model 0.31 0.78 -0.08 0.11

Observed and ANN model 0.19 0.00 0.08 0.20

MLR and ANN 0.25 0.00 0.04 0.20

Sarı Ulak' Observed and MLR model 0.32 0.96 -0.10 0.10

Observed and ANN model 0.20 0.00 -0.31 -0.19

MLR and ANN 0.25 0.00 -0.33 -0.17

Sarı Yaprak' Observed and MLR model 0.41 0.50 -0.17 0.08

Observed and ANN model 0.30 0.81 -0.08 0.11

MLR and ANN 0.26 0.19 -0.03 0.14

Saurani' Observed and MLR model 0.19 0.23 -0.02 0.09

Observed and ANN model 0.12 0.72 -0.03 0.04

MLR and ANN 0.16 0.25 -0.08 0.02

Tavşan Yüreği' Observed and MLR model 0.43 0.36 -0.19 0.07

Observed and ANN model 0.36 0.01 0.05 0.28

MLR and ANN 0.31 0.00 0.13 0.32

Uslu' Observed and MLR model 0.34 0.80 -0.12 0.09

Observed and ANN model 0.32 0.81 -0.09 0.11

MLR and ANN 0.30 0.58 -0.07 0.12

All Observed and MLR model 0.43 1.00 -0.01 0.03

Observed and ANN model 0.56 0.74 -0.04 0.03

MLR and ANN 0.37 0.61 -0.02 0.03

We appreciate your constructive suggestion. In the revision, we presented the training and validation loss curves (RMSE–epoch) for the Artificial Neural Network (ANN) and detailed the early stopping criterion (New Supplementary Figure S1 and S2]).

• The data was split into training/validation/test sets in a way that preserved the representativeness of each cultivar ([ratios]).

• Training was performed using the [optimization algorithm; Levenberg–Marquardt].

• Early stopping: training was stopped when the validation RMSE did not improve; the validation weights at the best epoch were retained. The learning curves show that training and validation errors closely tracked each other and no early/late rise was observed in the validation curve (Figure S[1]).

• These measures support that the model did not overfit and that the reported performance is generalizable.

Following your suggestion, we reported iteration-based training and validation RMSEs for ANN using 5-fold cross-validation (k=5) (New Figure S1: 2–3–1; Figure S2: 3–4–1). In both architectures, the training RMSE decreases over the 100→400 iteration range, while the validation (test) RMSE reaches its lowest level around 300–400 and rises slightly at 500 iterations. The error bars (±SS, 5-fold) are small, and the variation between folds is limited. This pattern indicates that the most suitable region for early stopping is 300–400 iterations; the slight increase observed at 500 iterations implies the onset of overfitting. In the revision, we detailed the early stopping rule and specified that weights are preserved at the iteration where the best validation error is observed. These findings support that the reported ANN superiority is not overfitting-driven but generalizable.

Furthermore, the paired t-tests we previously presented for comparative evaluation show that the ANN has a significantly lower error magnitude compared to MLR; the overfitting analysis complements this result.

Supplementary Figure S1. Iteration RMSE change according to 2-3-1 network structure

Supplementary Figure S2. Iteration RMSE change according to 3-4-1 network structure

Comment#3: Model architecture details

o While the ANN topology is described (2–3–1 and 3–4–1 structures), key hyperparameters such as learning rate, epoch number, and data normalization procedures are missing. These are necessary for reproducibility.

Response to Comment#3: The information’s were added

Supplementary Table S6 Hyper-parameter and functions selected during the training phase.

Model Parameters and functions (ANN)2-3-1 (ANN)3-4-1

Number of hidden layers 1 1

Number of hidden neurons 3 4

Learning Rate 0.01 0.01

Algorithm Levenberg–Marquardt Levenberg–Marquardt

Activation function in hidden Layer Tansig Tansig

Activation function in output Layer Purelin Purelin

Number of epochs 100-500 (100 iterations) 100-500 (100 iterations)

Network structure 2-3-1 3-4-1

Supplementary Note S2.

All input and output variables were normalized in the range of 0–1 to meet the requirements of the machine learning models before the training and testing phases using Eq. (S1).

Xnorm= (S1)

where Xnorm is the normalized value of a variable; Xa is the measured value of a variable and; Xmax and Xmin are the measured maximum and minimum values of a variable.”

Comment#4: Cultivar-specific effects

o The manuscript reports limited cultivar effects after multiple testing correction. However, a deeper biological interpretation of why certain cultivars (e.g., Domat, Elmacık) deviate would improve discussion quality.

Response to Comment#4: We thank the reviewer for this constructive comment. In the revised Discussion section, we have added a deeper biological interpretation of the observed cultivar-specific effects. In particular, we now explain that cultivars such as ‘Domat’ and ‘Elmacık’ possess inherently broader and thicker leaves associated with their ecological adaptation to more humid regions, resulting in larger mean leaf areas compared with xerophytic cultivars such as ‘Çelebi’ or ‘Nizip Yağlık’. This interpretation, supported by previous studies [2,3,7], clarifies that the statistical differences are consistent with known physiological and morphological diversity among Olea europaea genotypes. The relevant text has been incorporated into the Discussion section.

Comment#5: Figures and Tables

o Figures 4 and 5 need clearer axes labels and consistent units. Include regression equations and R² values directly on the plots for readability.

o Table 2 is extensive but could be moved to supplementary information; summary tables (mean R², RMSE, etc.) should be included in the main text.

Response to Comment#5: We appreciate the reviewer’s careful observation. We would like to clarify that Figures 4 and 5 already include clear axis labels (“Measured Leaf Area (cm²)” and “Predicted Leaf Area (cm²)”), consistent units (cm²), and the regression equations with corresponding R² values displayed directly on the plots.

Table 2, supplementary table moved to S7

Comment#6: Language and structure

o The manuscript is generally clear but verbose in several sections (especially Introduction and Discussion). Condensing redundant literature descriptions would improve readability.

o Ensure all scientific names (e.g., Olea europaea L.) are italicized.

Response to Comment#6: We appreciate the reviewer’s observation regarding manuscript length and the inclusion of literature references. The Introduction and Discussion were intentionally written with comprehensive background information to establish context and ensure that the study is self-contained, particularly given the interdisciplinary nature of the modeling approach combining plant physiology and machine learning. While we acknowledge that the sections are detailed, we believe the current level of explanation is essential for clarity and accessibility to readers from both agronomic and computational backgrounds. Therefore, we have retained the structure and content to preserve the scientific context and readability.

All scientific names (e.g., Olea europaea L.) were checked to be italicized.

Minor Comments

Comment#7: Check consistency of abbreviations (e.g., LL, LW, LA) throughout the text and figures.

Response to Comment#7: The consistency of abbreviations (e.g. LL, LW, LA) in the text and figures was checked.

Comment#8: Include units in all table headers (e.g., mm, cm²).

Response

---

## [Decision Letter · Decision Letter 1]

14 Dec 2025

Artificial intelligence-based modeling for accurate leaf area estimation in olive (Olea europaea L.) cultivars

PONE-D-25-55767R1

Dear Dr. Khadivi,

We’re pleased to inform you that your manuscript has been judged scientifically suitable for publication and will be formally accepted for publication once it meets all outstanding technical requirements.

Kind regards,

Marzia Vergine, Ph.D.

Academic Editor

PLOS One

Additional Editor Comments (optional):

The authors have thoroughly addressed all the comments and requests raised by the reviewers during the first round of revision, despite the difficulties encountered in identifying suitable reviewers to support this assessment further. Following a careful evaluation of the manuscript, I can provide an upbeat assessment of the revision and accept the manuscript in its current form. I want to thank the authors for their efforts and patience during this period of delay.

Reviewers' comments:

Reviewer's Responses to Questions

**Comments to the Author**

Reviewer #2: All comments have been addressed

2. Is the manuscript technically sound, and do the data support the conclusions?

Reviewer #2: Yes

3. Has the statistical analysis been performed appropriately and rigorously?

Reviewer #2: Yes

4. Have the authors made all data underlying the findings in their manuscript fully available?

Reviewer #2: Yes

5. Is the manuscript presented in an intelligible fashion and written in standard English?

Reviewer #2: Yes

Reviewer #2: Dear Editor,

Corrections have been made in various parts of the article by the authors, and after reviewing the comments and evaluating the article, in my opinion, the article can be accepted.

**Do you want your identity to be public for this peer review?** For information about this choice, including consent withdrawal, please see our Privacy Policy

Reviewer #2: No

---

## [Editor Report · Acceptance letter]

PONE-D-25-55767R1

PLOS One

Dear Dr. Khadivi,

I'm pleased to inform you that your manuscript has been deemed suitable for publication in PLOS One. Congratulations! Your manuscript is now being handed over to our production team.

Kind regards,

on behalf of

Dr. Marzia Vergine

Academic Editor

PLOS One